# Radiosensitivity of Cancer Cells Is Regulated by Translationally Controlled Tumor Protein

**DOI:** 10.3390/cancers11030386

**Published:** 2019-03-19

**Authors:** Jiwon Jung, Ji-Sun Lee, Yun-Sil Lee, Kyunglim Lee

**Affiliations:** Graduate School of Pharmaceutical Sciences, College of Pharmacy, Ewha Womans University, Seoul 03760, Korea; major87@hanmail.net (J.J.); leejisun78@hanmail.net (J.-S.L.)

**Keywords:** TCTP, radioresistance, p53

## Abstract

Translationally controlled tumor protein (TCTP) is a ubiquitous multifunctional protein that is essential for cell survival. This study reveals that the regulation of radiosensitivity of cancer cells is yet another function of TCTP. The relationship between endogenous TCTP levels and sensitivity to radiation was examined in breast cancer cell lines (T47D, MDA-MB-231, and MCF7) and lung cancer cells lines (A549, H1299, and H460). Cancer cells with high expression levels of TCTP were more resistant to radiation. TCTP overexpression inhibited radiation-induced cell death, while silencing TCTP led to an increase in radiosensitivity. DNA damage in the irradiated TCTP-silenced A549 cells was greater than in irradiated control shRNA-transfected A549 cells. p53, a well-known reciprocal regulator of TCTP, was increased in irradiated TCTP down-regulated A549 cells. Moreover, introduction of p53 siRNA in TCTP knocked-down A549 cells abrogated the increased radiosensitivity induced by TCTP knockdown. An in vivo xenograft study also confirmed enhanced radiosensitivity in TCTP down-regulated A549 cells. These findings suggest that TCTP has the potential to serve as a therapeutic target to overcome radiation resistance in cancer, a major problem for the effective treatment of cancers.

## 1. Introduction

Radiation therapy (RT) is an important treatment option for cancer therapy, where it can be used alone or in combination with chemotherapies. Over 50% of cancer patients receive RT during the course of their illness and 40% of those cured have received RT as part of their treatment [1]. However, acquired radioresistance of tumor cells during radiation treatment can limit the usefulness of RT [2]. The mechanism behind resistance of cancer cells to RT remains largely elusive. Therefore, controlling the radioresistance of cancer cells is important for successful RT.

Translationally controlled tumor protein (TCTP) is a highly conserved protein present in all eukaryotic cells. TCTP has been implicated in important cellular process, such as cell growth, apoptosis, DNA damage repair, tumorigenesis, cancer progression, and protection against various stress conditions [3]. TCTP expression is higher in cancer cells compared to normal cells [4], and the higher TCTP statuses in breast [5], hepatocellular [6], ovarian [7], and brain [8] cancers are associated with poor prognosis. TCTP appears to be one of important factors for tumor reversion, in which v-Src-transfected NIH3T3 cells returned to their normal phenotype after TCTP down-regulation [9]. Glioma patients with lower TCTP levels showed better one-year survival rates after RT with adjuvant temozolomide treatment, suggesting that low TCTP level in cancer may correlate with enhanced radiosensitivity [10].

One of the main mechanisms of RT is the induction of DNA damage directly by generating DNA double-strand breaks (DSB), or indirectly by generating reactive oxygen species (ROS) [11]. TCTP was known to be upregulated in an ataxia-telangiectasia mutated kinase (ATM)-dependent manner, providing protection against direct DNA damage induced by low-dose γ-ray [12] and against oxidative stress in MDA-MB-231 cells [13]. From these reports, we can infer that TCTP may play a critical role in radiation resistance of cancer cells, but the exact mechanism underlying the role of TCTP in this regard has not been addressed yet.

In this study, we investigated TCTP association with the radiosensitivity of cancer cells and the underlying molecular mechanisms possibly involved. We found that the endogenous TCTP levels determine the radiosensitivity of cancer cells and that TCTP down-regulation increases the radiosensitivity of cancer cells, suggesting a new therapeutic avenue for preventing or decreasing the radioresistance of cancer.

## 2. Results

### 2.1. TCTP Expression Inversely Correlates with Radiosensitivity of Cancer Cells

To understand the role of TCTP in the radioresistance of cancer cells, we first investigated the TCTP levels in breast cancer cell lines (T47D, MDA-MB-231, and MCF7) and lung cancer cell lines (A549, H1299, and H460), using Western blotting. In comparing breast cancer cell lines, we found that TCTP expression was relatively lower in MCF7 cells than in T47D and MDA-MB-231 cells (Figure 1a). Next, the clonogenic formation assay was performed and the survival fraction of MCF7 cells at 2 Gray (Gy), which had low TCTP expression, was 0.423 ± 0.02. This was significantly lower than survival fractions of T47D (0.670 ± 0.11) and MDA-MB-231 (0.626 ± 0.06) cells (Figure 1b,c). Annexin V/PI assay showed that the cell death (Annexin V positive, PI positive, and double positive population) of MCF7 cells (25.71 ± 1.32%) was significantly higher than the cell death of MDA-MB-231 (15.81 ± 1.10%) and T47D cells (14.71 ± 0.94%) (Figure 1d,e), which accords with the survival fraction results.

We conducted similar comparisons with the three lung cancer cell lines. TCTP expression level of A549 cells was significantly higher compared to H460 and H1299 cells (Figure 2a). Comparative clonogenic formation assays revealed that survival fraction of A549 cells at 2 Gy was 0.896 ± 0.03, which was significantly higher than that of H1299 (0.639 ± 0.09) and H460 (0.518 ± 0.06) cells (Figure 2b). Consistent with clonogenic formation assay results, the cell death after γ-radiation treatment was significantly lower in A549 cells (11.94 ± 1.66) compared to H1299 (22.1 ± 2.93%) and H460 (28.36 ± 3.9%) cells (Figure 2c,d). The cancer cells with relatively high TCTP levels clearly showed more resistance to γ-radiation than the cancer cells with relatively low TCTP levels, confirming that high TCTP expression levels decreased the radiosensitivity of both breast cancer and lung cancer cell lines.

### 2.2. TCTP Is Involved in Radioresistance of Cancer Cells

We next investigated whether the expression of TCTP regulates the sensitivity to irradiation in breast and lung cancer cells. MCF7 cells were transfected with TCTP-3XFLAG and TCTP overexpression was confirmed by Western blotting (Figure 3a). They were then subjected to various doses of γ-radiation and incubated for 14 days to evaluate colony-forming ability. TCTP overexpression significantly increased the survival fraction of irradiated MCF7 cells (Figure 3b). TCTP-overexpressing MCF7 cells were treated with 10 Gy of γ-radiation and the cell death was measured after 48 h using flow cytometry. The cell death level of TCTP-3XFLAG-transfected MCF7 cells (14.73 ± 1.83%) was significantly lower than cell death rate of p3XFLAG-transfected MCF7 cells (23.10 ± 1.22%) (Figure 3c,d). H460 cells were also transfected with TCTP-3XFLAG and the cell death level was measured after irradiation (Appendix A). TCTP-overexpressing H460 cells showed a significant decrease in cell death rate (21.27% ± 2.42) compared to p3XFLAG-transfected H460 cells (28.18% ± 1.45) (Appendix A). These results indicate that exogenous expression of TCTP increased the sensitivity to irradiation. Then, TCTP was knock-downed using shRNA in A549 cells (Figure 3e). In contrast to TCTP-overexpressed MCF7 cells, TCTP down-regulated A549 cells showed decreased survival fraction (Figure 3f and Appendix A). The cell death percentage of γ-radiation-treated TCTP down-regulated A549 cells (16.95 ± 0.67%) was significantly increased from control shRNA-transfected A549 cells (9.36 ± 0.44%) (Figure 3g,h). We further generated A549 stable cells that expressed shRNA for TCTP using the pLKO.1 lentiviral vector system. Briefly, A549 cells were infected with TCTP-specific shRNA-expressing lentivirus (shTCTP) or control lentivirus (shCont) and clones were obtained, in which TCTP was significantly and stably down-regulated (Appendix A). γ-radiation-treated stable cells with suppressed TCTP expression showed increased dead cell population (13.68 ± 0.78%) compared to control cells (19.85 ± 0.53%) (Appendix A). Collectively, these results show that breast and lung cancer cells’ sensitivity is regulated in a TCTP-dependent manner.

### 2.3. TCTP Knockdown Potentiates Radiation-Induced DNA Damage

One of the main mechanisms of cell death induced by γ-radiation is DNA damage [14]. Since it is well known that TCTP plays an important role in DNA damage sensing and repair [12], we examined whether the radiation-induced DNA damage was increased after TCTP shRNA transfection. Since radiation phosphorylates H2AX at constant rate of γ-H2AX per DSB [15], we determined DNA damage caused by γ-radiation in A549 cells introduced with control or TCTP shRNA by measuring γ-H2AX level. Western blotting assay showed that TCTP down-regulated A549 cells treated with γ-radiation showed the highest level of γ-H2AX compared to the other groups (Figure 4a). An increase in phosphorylation of H2AX was also observed in γ-radiation-treated cells with stable TCTP knocked-down A549 cells (Appendix A). Also, comet assay was used to confirm the DNA damage level of TCTP shRNA-transfected A549 cells. The relative olive tail moments of TCTP down-regulated A549 cells, radiation only-treated A549 cells, and radiation-treated TCTP down-regulated A549 cells were 1.37 ± 0.7, 1.50 ± 0.07, and 2.61 ± 0.10, respectively. TCTP down-regulated A549 cells and γ-radiation-treated A549 cells showed a slight increase in olive tail moment, while the γ-radiation-treated TCTP down-regulated A549 cells showed the highest olive tail moment (Figure 4b). The increased γ-H2AX level and olive tail moment suggest that TCTP knockdown sensitizes cancer cells to radiation-induced DNA damage.

### 2.4. p53 Is Involved in TCTP-Mediated Radioresistance of A549 Cells

It is known that TCTP binds to mouse double minute 2 homolog (MDM2) and inhibits MDM2 auto-ubiquitination, which promotes MDM2-mediated p53 degradation [5]. To understand the molecular mechanism behind TCTP-induced radioresistance, the expression levels of p53 and MDM2 after γ-radiation were measured in TCTP knocked-down A549 cells. The p53 level was significantly elevated in irradiated TCTP shRNA-transfected A549 cells (2.38 ± 0.22) compared to irradiated control shRNA-transfected A549 cells (1.52 ± 0.62), which was in reciprocal manner with the expression level of TCTP. Also, MDM2 stabilization in response to γ-radiation was less pronounced in TCTP down-regulated cells (2.04 ± 0.72) compared to the control cells (3.98 ± 1.60) (Figure 5a). Increased p53 levels in TCTP shRNA-transfected γ-irradiated A549 cells suggest that p53 may be involved in TCTP knockdown-induced radiosensitization. Since p53 induces G1 phase arrest in the cell cycle by inducing p21 [16] and consequent inhibition of cyclin E/Cdk2 and cyclin D/Cdk4 complexes [17] and activated p53 also causes G2/M phase arrest by induction of 14-3-3σ or by decreasing cyclin B1 [18,19], we performed cell cycle analysis in TCTP knockdown and control A549 cells (Figure 5b,c). Radiation treatment resulted in G2/M phase arrest in control A549 cells (42.02 ± 2.51%) compared to the untreated cells (22.8 ± 0.92%). When TCTP was down-regulated, cell population was significantly more arrested in G2/M phase after radiation treatment (51.09 ± 5.04 %) when compared to radiation-treated control cells. This result suggests the possibility of TCTP-mediated radioresistance through the function of p53 in inducing cell cycle arrest in G2/M phase and/or delay in DNA repair system. This result is consistent with the previous reports that TCTP serves a role in regulating the magnitude of the G2 phase arrest by radiation [12]. In order to determine whether p53 is involved in TCTP-mediated radioresistance, p53 siRNA was introduced to TCTP down-regulated A549 cells and cell death was measured using Annexin V/PI assay (Figure 5d,e). The increased cell death percentage of TCTP-depleted A549 cells (18.28 ± 1.00%) was decreased when p53 expression was inhibited (11.39 ± 0.57%) (Figure 5f), suggesting there was p53 involvement in radiosensitization caused by TCTP knockdown in A549 cells.

### 2.5. TCTP Knockdown Enhances Radiosensitivity in Tumor Xenograft Mouse Model

In order to confirm TCTP-dependent radiosensitivty in vivo, we analyzed the effect of γ-radiation on tumor growth in mice bearing A549-derived tumors. We injected control and stable TCTP knocked-down A549 cells into the right thigh of 6-week-old, female, Balb/c nude mice. When the tumor volume reached approximately 100 mm^3^, the mice were treated with 8 Gy of γ-radiation and the growth of the tumors were assessed by digital caliper. Since our previous results showed that there was no significant difference in radiosensitivities of untransfected A549 cell and control shRNA introduced A549 cells, we used control shRNA-introduced A549 cells (shCont-A549) as a control. The tumor growth of shTCTP-A549 cells treated with γ-radiation were delayed compared to the tumor growth of shCont-A549 cells exposed to the same dose of γ-radiation (Figure 6a). Moreover, the average tumor growth rate of shTCTP-A549 cells treated with γ-radiation (4.946 ± 1.73 mm^3^/days) was significantly lower than the growth rate of shCont-A549 cells treated with γ-radiation (12.47 ± 2.0 mm^3^/days) (Figure 6b). The reciprocal regulation of TCTP and p53 was also observed in tumors of shCont-A549 and shTCTP-A549 xenografted tumors. Down-regulation of TCTP and up-regulation of p53 were observed in tumors derived from shTCTP-A549 by Western blot analysis (Figure 6c). Since it is known that TCTP enhances cell growth and its down-regulation decreases Ki-67 positive cells [10], we performed immunohistochemical studies using the anti-Ki-67 antibody and determined the percentage of Ki-67 positive cells. The percentage of Ki-67 positive cells in shCont-A549 group (39.01% ± 5.13) was higher than the percentage of the shTCTP-A549 group (24.16% ± 3.91) as well as the percentage of the irradiated shCont-A549 group (22.67% ± 3.54). The percentage of Ki-67 positive cells in the irradiated shTCTP-A549 group (11.05% ± 2.72) was the lowest (*p* < 0.05) (Figure 6d). Using in vivo models, we confirmed that TCTP inhibition combined with radiation treatment enhanced the anti-tumor effect compared to radiation treatment alone.

## 3. Discussion

TCTP is a ubiquitously expressed protein that participates in numerous cellular processes. TCTP’s association with cancer and its higher expression in cancer cell lines has been well established [4]. High TCTP expression levels in breast cancer cells correlates with poor differentiation, high proliferation, and low or negative estrogen receptor expression, all features of clinical and pathological aggressiveness [5]. TCTP has been reported to positively correlate with clinicopathological features of glioma [10]. In addition, TCTP expression correlates with poor survival, high pathological grades, and M stage classification in tumor, node, metastasis (TNM) stage in colorectal cancer patients [20]; TCTP is known to be increased, inducing resistance to 5-fluorouracil treatment in colon cancer [21].

TCTP’s role has been implicated from regulating tumorigenesis to resistance against chemotherapeutic agents in various cancers, but its involvement in radiosensitivity of cancer cells is not known. Our study is the first to establish the relationship between TCTP expression and the radioresistance of cancer cells in vitro and in vivo. We examined the relationship between TCTP expression and radioresistance in breast and lung cancer cells and discovered that cancer cells with high TCTP expression levels showed greater resistance to radiation in both breast (Figure 1) and lung (Figure 2) cancer cells.

The functional analysis through gain- or loss- of function studies showed that TCTP overexpression increased clonogenic survival fraction and decreased cell death percentage, whereas TCTP knockdown using TCTP shRNA showed the opposite effects (Figure 3). It is known that TCTP overexpression gives resistance to chemotherapy such as etoposide and Taxol treatment in HeLa by inhibiting mitochondrial membrane damage, cytochrome c release, and activation of caspase-9 and -3 [22]. Also, TCTP down-regulation significantly inhibited proliferation and invasion and induced apoptosis in glioma cells [23] and inhibition of TCTP led to the reduction of cell viability of prostate cancer cells [24]. Our results suggest that TCTP regulation not only determines aggressiveness of cancer, but also may be involved with the radioresistance of cancer cells.

One of the main mechanisms of radiation-induced cell death is through DNA damage. Thus, we measured the DNA damage in TCTP down-regulated A549 cells using γ-H2AX as biomarker. H2AX is a nucleosomal protein, which is phosphorylated on Serine 139 site in response to DNA DSB [25]. Irradiated TCTP down-regulated A549 cells showed the greatest induction of γ-H2AX compared to rest of the groups (Figure 4a). Persistence of γ-H2AX after radiation is reported as a marker for radiosensitivity of cancer cells [26] and γ-H2AX loss is relatively faster in radioresistant cell lines [27]. Our study showed prolonged DNA damage in TCTP-depleted A549 cells, accompanied with the loss of radioresistance (Figure 3 and Figure 4). To confirm the DNA damage seen in TCTP down-regulated A549 cells, comet assay was performed under neutral pH conditions. Agreeing with the γ-H2AX measurement, comet assay showed that radiation-treated TCTP down-regulated A549 cells had the highest olive tail moment (Figure 4b). Zhang et al. reported that TCTP is up-regulated rapidly in an ATM- and DNA-dependent protein kinase (DNA-PK)-dependent manner when exposed to low doses of γ-ray [12], indicating possible regulation of TCTP by ATM and DNA-PKcs and its participation in repair of DSB. In preliminary experiments, phosphorylation of ATM in TCTP down-regulated A549 cells was measured and low phosphorylation levels of ATM were observed after radiation. Even though much still needs to be learned, the findings already accumulated from our studies and those of others suggest that TCTP may participate in repair of DSB and function in DNA damage response.

Radiation induces DNA damage directly by targeting DNA strands or indirectly by generating reactive oxygen species (ROS) in cells [14,28], which leads to DNA breaks and massive phosphorylation of H2AX [29]. Lucibello et al. reported that TCTP was up-regulated in cancer cells which survived mild oxidative stress and that sensitivity to oxidative stress was strongly enhanced when high levels of TCTP were down-regulated in the cells [13]. In addition, recent reports suggest that TCTP acts like an antioxidant protein and upregulates peroxiredoxin in transgenic mice [30,31]. The persistence of γ-H2AX expression and increased olive tail moment in TCTP down-regulated A549 cells after radiation may be the result of DNA damage accumulation due to the absence of protection from TCTP against ROS (Figure 4).

The tumor suppressor p53 plays crucial role in growth arrest, apoptosis, and DNA repair [32]. Since p53 level is increased in cells with DNA damage [33] and p53 and TCTP work as reciprocal regulators [5], we used TCTP down-regulated A549 cells to observe the changes in p53 and MDM2 levels after radiation. MDM2 induction in response to radiation was less pronounced in TCTP down-regulated A549 cells than in control cells, whereas p53 induction was more pronounced in TCTP down-regulated A549 cells than in control cells (Figure 5a). Chen et al. discovered that after little DNA damage, p53 was slightly increased, but after great DNA damage, p53 was greatly increased, activating apoptosis of cells [34]. Also, it was reported that persistent γ-H2AX level in DNA-damaged cells activates ATM–p53 mediated apoptosis [35]. The increased p53 level, as seen in Figure 5, may be a result of high DNA damage as indicated by prolonged γ-H2AX, shown in Figure 4.

The damaged DNA phosphorylates γ-H2AX and promotes activation and recruitment of other DNA response proteins, including p53 [36]. It is well known that MDM2 is the p53 binding protein that ubiquitinates p53 and helps p53 degradation [37]. Amson et al. described how TCTP and p53 are regulated in reciprocal manner; TCTP regulates p53 via MDM2 stabilization and p53 regulates TCTP by binding to its promoter. After whole body irradiation, increase in cell death of thymocyte of Tctp^+/−^ mice but not of Tctp^+/−^; Trp53^−/−^ mice was observed [5]. In accordance with the previously reported studies, we discovered that TCTP down-regulation caused up-regulation of p53 during radiation treatment through MDM2 destabilization. When p53 siRNA was introduced to TCTP silenced A549 cells, the radiosensitizing effect of TCTP down-regulation was abrogated, suggesting p53 involvement (Figure 5f).

In addition, in vivo studies verified that TCTP down-regulation sensitized cancer cells to radiation. The growth of tumor volume and average tumor growth rate of shTCTP-A549 after radiation were smaller compared to tumors derived from shCont-A549 on the day of sacrifice (Figure 6). Increased p53 level in shTCTP-A549 was more potentiated by radiation, which is in agreement with the in vitro data (Figure 5a). Also, immunohistochemical studies using Ki-67 showed that the cell proliferation rate was the lowest in irradiated shTCTP-A549 (shTCTP+IR) tumors, which is in accordance with average tumor growth rate. These results are consistent with reports of decreased tumor cell proliferative capability in breast cancer, prostate cancer, lung cancer, and squamous cell carcinoma after TCTP down-regulation [24,38,39,40]. Along our finding that describes increased cell death caused by p53 induction after TCTP down-regulation, there has been a report that TCTP overexpression increases glioma cell proliferation and tumor growth of xenograft models by enhancing β-catenin association with TCF-4, thereby increasing TCF-4/β-catenin transcriptional activity [10]. Thus, from these findings, we can speculate that TCTP modulation can regulate radiosensitivity in cancer cells in vitro and in vivo.

Intracellular TCTP levels determine radiosensitivity in cancer cells and TCTP down-regulation has sensitized A549 cells to radiation by increasing p53 induction and DNA damage. Our study suggests the possibility of using the concept of down-regulation of TCTP as a new biological modifier that increases responsiveness of cancer cells to radiation without any undesirable side effects.

## 4. Materials and Methods

### 4.1. Cell Lines and Culture Conditions

All cell lines were obtained from American Type Culture Collection (ATCC, Manassas, VA, USA). MDA-MB-231, T47D, A549, H1299, and H460 were maintained at 37 °C in 5% CO_2_ in Roswell Park Memorial Institute (RPMI) 1640 medium (Invitrogen, Carlsbad, CA, USA) supplemented with 10% (v/v) fetal bovine serum (FBS), 100 units/mL penicillin, and 100 units/mL streptomycin. MCF7 was grown in Dulbecco’s Modified Eagle’s Medium (DMEM) (Invitrogen), supplemented with 10% (v/v) fetal bovine serum (FBS), 100 units/mL penicillin, and 100 units/mL streptomycin.

### 4.2. Antibodies

TCTP (Santa Cruz Biotechnology, sc-8334) and MDM2 (Santa Cruz Biotechnology, sc-813) were purchased from Santa Cruz Biotechnology (Santa Cruz, CA, USA). β-actin (Cell Signaling Technology (Beverly, MA, USA) 4970), p53 (Cell Signaling Technology, 9282), γ-H2AX (Cell Signaling Technology, 2577), and H2AX (Cell Signaling Technology, 2595) were purchased from Cell Signaling Technology.

### 4.3. Clonogenic Formation Assays

A total of 500–4000 cells were seeded in triplicate 60 mm dishes and irradiated with various doses of γ-radiation in a ^137^Cs unit at room temperature. The cells were then cultured in an incubator containing 5% CO_2_ at 37 °C for 14 to 16 days. Individual colonies (>50 per colony) were fixed and stained with a solution containing 5% crystal violet in methanol for 2 h and counted manually. Each experiment was done in triplicate.

### 4.4. Irradiation

Cells were exposed to γ-ray with a ^137^Cs-ray source (Atomic Energy of Canada, Mississauga, ON, Canada) with a dose rate of 3.18 Gy/min.

### 4.5. Transfection

The construct for intracellular synthesis of TCTP shRNA was made in pSUPER vector (OligoEngine, Seattle, WA, USA) using two oligos, forward (5′-GTCC CCAGGTACCGAAAGCACAGTATTCAAGAGATACTGTGCTTTCGGTACCTTTTTTGGAAA-3′) and reverse (5′-AGCTTTTCCAAAAAAGGTACCGAAAGCA CAGTATCTCTTGAATACTGTGCTTTCGGTACCTG GG-3′). The annealed oligos were cloned into the vector linearized with BalII and HindIII. The functional shRNA corresponding to 5′-AGGTACCGAAAGCACAGTA-3′ of human TCTP gene was synthesized inside mammalian cells transfected with TCTP shRNA construct [41].

For TCTP overexpression, human TCTP/p3X FLAG-CMV-14 vector was generated as described with minor modifications [42]. Briefly, using primers with EcoRI and BamHI sites, human TCTP insert was made by PCR amplification. Insert DNA and p3XFLAGCMV-14 empty vector were excised by BamHI and EcoRI restriction enzymes and sticky-end ligated. The plasmid was confirmed by DNA sequencing. The cells were seeded and, a day later, the cells were transfected with these vectors using Lipofectamine 2000 (Invitrogen) or Welfect Q-Gold (Welgene, Daegu, South Korea), following manufacturer’s protocol.

### 4.6. Generation of Stable Cell Lines by Lentiviral Transduction

A stable cell line that showed TCTP knockdown was generated as previously described [43], with minor modification. Briefly, HEK-293FT cells were transfected with pLKO.1 (shCont-A549) or pLKO.1-shTCTP (shTCTP-A549) vectors with pVSV-G and Δ8.9, using Lipofectamine 2000 (Invitrogen). The supernatants of cell culture containing viruses were harvested for infection. The virus-containing media was supplemented with polybrene (Sigma, St. Louis, MO, USA), then filtered through a 0.45 μm syringe-driven to remove debris. The filtered media was added to the culture of A549. After incubation with virus particles, infected cells were selected through puromycin treatment (Sigma).

### 4.7. Immunoblotting

Samples were immunoblotted as previously described with modifications [44]. Cells were harvested using scrapper and lysed in lysis buffer (50 mM Tris-Hcl (pH 7.4), 150 mM NaCl, 1 mM EDTA, 2 mM Na_3_VO_4_, 1 mM NaF, 0.25% deoxycholate, 1% Triton X-100, and protease inhibitor cocktail tablet (Roche, Mannheim, Germany)) for 30 min. The lysates were centrifuged at 12,000× *g* for 20 min at 4 °C. After determining the protein content in the soluble cytosolic fraction content by Bradford protein assay (Bio-Rad, Hercules, CA, USA), the samples were diluted in SDS sample buffer (1 M Tris-HCl (pH 6.8), 10% SDS, 10% glycerol, 0.2% bromophenol blue and 2% β-mercaptoethanol). The samples were separated at 10% or 12% SDS-PAGE gel, and transferred to PVDF membrane (Bio-Rad, CA, USA). The membrane was probed with antibodies, followed by probing with an appropriate secondary antibody (Bio-Rad). Detected proteins were visualized by enhanced chemiluminescence (ECL) and UV Products Imaging system, LAS 3000 (Fuji, Japan).

### 4.8. Annexin V PI Apoptosis Assay

The cells were prepared and stained with the Annexin V-FITC Apoptosis Detection Kit (BD Pharmingen, San Jose, CA, USA) according to the manufacturer’s instructions. Briefly, 3 × 10^5^ cells were seeded on 60 mm culture dish and treated with the indicated dose of γ-radiation. After 48 h, cells were harvested by trypsinization, pelleted by centrifugation, resuspended in 1 × Annexin V binding buffer (BD Pharmingen), and treated with Annexin V-FITC and PI (BD Pharmingen) as instructed by manufacturer’s protocol. Samples were analyzed by flow cytometry (FACS Calibur, BD, San Jose, CA, USA) using FL1 and FL3 channels. Flow cytometric analysis was performed using CellQuest analysis software (BD Biosciences, San Jose, CA, USA).

### 4.9. Comet Assay

The comet assays were performed under neutral conditions as previously described [45], with minor modifications. Briefly, A549 cell suspension was mixed with low-melting agarose at 1 × 10^4^ cells/mL and evenly pipetted onto the microscope slides. The slides were maintained at 4 °C for 10 min to solidify. The slides were immersed in chilled lysis solution (Trevigen, Gaithersburg, MD, USA) at 4 °C for 2 h in the dark. The slides were immersed in TBE buffer (45 mM Tris-borate, 1 mM EDTA) for 10 min. Electrophoresis was then performed at 25 V, 150 mA, for 20 min. The slides were washed with distilled water, fixed at 70% EtOH for 5 min, and stained with 20 μL SYBR (Invitrogen). Images of comets were captured under fluorescence microscope at 100× magnification. For each sample, a minimum of 50 comets were obtained and the olive tail moment (tail DNA (%) × (tail mean − head mean)) was analyzed using the Comet Assay Software (Komet 5.5, Andor technologies, Abingdon, UK).

### 4.10. Cell Cycle Analysis

In a 60 mm culture dish, 3 × 10^5^ cells were seeded and exposed to γ-radiation. 24 h after, cells were trypsinized, washed with PBS, fixed in 70% ethanol for 30 min at 4 °C, and washed with PBS twice. After RNase treatment (0.5 mg/mL in PBS) at 37 °C for 1 h, cells were stained with propidium iodide and analyzed using flow cytometry (FACS Calibur, BD, San Jose, CA, USA). Flow cytometric analysis was performed using CellQuest analysis software.

### 4.11. Experimental Animals

All animal studies were conducted in accordance with IACUC guidelines and were approved by the IACUC committee at Ewha Womans University (Approval ID: 14-097). The animals were housed under pathogen-free conditions and maintained at a controlled temperature and humidity. Standard diet (Cargill Agri Prunia, Seongnam, Korea) and water were supplied ad libitum.

### 4.12. Xenograft

Stably transfected A549 cells (5 × 10^6^ cells in 100 μL PBS) were subcutaneously injected into right thigh of 6-week-old, female, Balb/c Nude mice. When the tumor reached a mean tumor volume of 100 mm^3^, the mice were randomly assigned to 4 groups: 0 Gy and 8 Gy for tumors derived from stably transfected shCont-A549 cells and shTCTP-A549 cells, respectively. Mice assigned to the radiation treatment group were exposed to radiation under anesthesia (30 mg/kg Zoletil and 10 mg/kg Rompun) using X-Rad 320 X-ray irradiator (Precision X-ray, North Branford, CT, USA) at a dose rate of 1.5 Gy/min (260 kV, 10 mA). The radiation field size for localized tumor exposure was 30 × 50 mm. The tumor size was measured for 25 days and animals were sacrificed after 25 days. Tumor volume (V) was calculated as V = (length/2) × (width^2^). The tumor growth rate (U) was calculated as previously described with minor modification [46]. The tumor growth rate was calculated using the equation U = (V of end point − V of initial point)/tumor-bearing time (days).

### 4.13. Immunohistochemistry

Immunohistochemistry was performed as previously described with minor modifications [47]. Briefly, tissue samples were fixed in 10% formalin for 24 h, dehydrated in ethanol, and embedded in paraffin. Sections were cut on a rotary microtome to 5 μm and mounted on glass slides. After deparaffinization and hydration, the tissue sections were boiled for 20 min in sodium citrate buffer, pH 6.0, to retrieve the epitope. Endogenous peroxidase was blocked by incubation with 0.5% hydrogen peroxide for 15 min. After blocking in 5% BSA in PBS for 30 min at room temperature, the samples were incubated for 1 h at room temperature with the Ki-67 antibody (M7240, Dako, Santa Clara, CA, USA) 1:50) diluted in 0.5% BSA in PBS. Sections were washed and treated with HRP conjugate, and stained with DAB solution. Stained sections were analyzed under a microscope (Axio Scope, A1, Zeiss, Oberkochen, Germany) and representative photographs were taken with a digital camera.

The Ki67 proliferation index was determined by counting cells with positive nuclear staining in the representative field, as previously described with modifications [48]. Up to six representative fields containing an average of 200 cells were captured for each tumor group and cells with positive nuclear staining were manually counted by three different people.

### 4.14. Statistical Analysis

Data are presented as means and their standard errors. Data were analyzed using GraphPad Prisms 5 software (GraphPad Software Inc., San Diego, CA, USA). Statistical significance was determined using Student’s *t*-test (two samples) or ANOVA (multiple comparisons). A *p* value < 0.05 was considered statistically significant.

## 5. Conclusions

In conclusion, we identified the function of TCTP in regulating radiosensitivty in breast and lung cancer cell lines through gain- or loss-of-function studies. From this research, TCTP levels of cancer cells showed correlation with radioresistance and decreased expression of TCTP sensitized cancer cells to γ-radiation by increasing p53 expression and accumulation of prolonged DNA damage. Thus, our findings suggest TCTP regulation is a possible target to overcome the limited therapeutic efficacy of radiation therapy and a new therapeutic target for preventing the radioresistance of cancer.

## Figures and Tables

**Figure 1 cancers-11-00386-f001:**
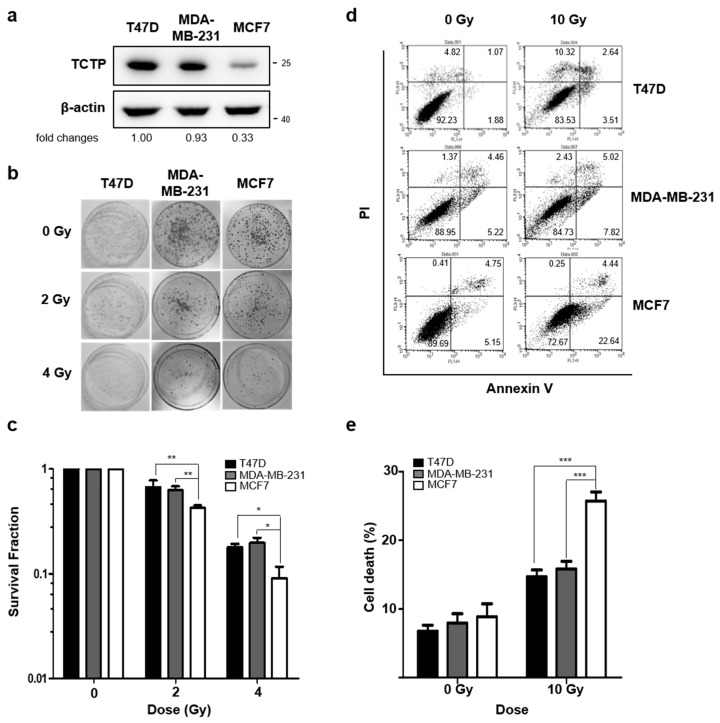
Translationally controlled tumor protein (TCTP) expression inversely correlates with sensitivity to γ-radiation in breast cancer cells. (**a**) TCTP expression of breast cancer cell lines were determined by Western blotting (*n* = 4). The cropped blots are used in the figure and full-length blots are presented in Appendix A. The relative level of TCTP in comparison to β-actin is indicated below each immunoblot image. (**b**,**c**) The cells were treated with different doses of γ-radiation, and the survival fraction was determined by a clonogenic formation assay (*n* = 3). (**b**) Representative image of clonogenic formation. (**c**) Survival fraction relative to each untreated group was calculated and shown in the graph. (**d**,**e**) Cells were treated with γ-radiation of 10 Gy and dead cell populations were determined by flow cytometry after staining with Annexin V and PI (*n* = 4). (**d**) Representative image of PI-Annexin V double staining examined in breast cancer cells and (**e**) graph of dead cells is shown. Bars represent the means ± SEM. * *p* < 0.05, ** *p* < 0.01, *** *p* < 0.001 by two-way analysis of variance.

**Figure 2 cancers-11-00386-f002:**
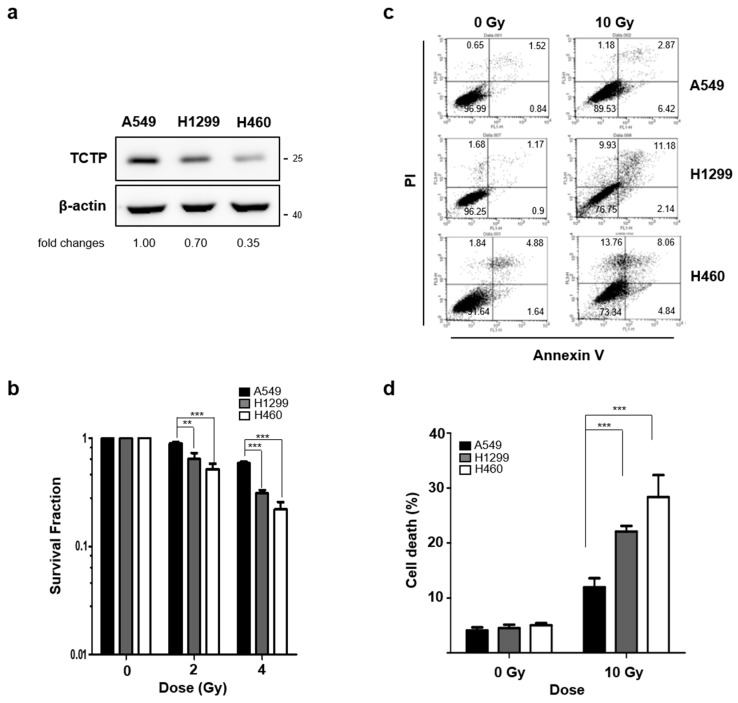
TCTP expression inversely correlates with sensitivity to γ-radiation in lung cancer cells. (**a**) TCTP expression of indicated lung cancer cell lines were determined by western blot analysis (*n* = 4). The cropped blots are used in the figure, and full-length blots are presented in Appendix A. The relative level of TCTP in comparison to β-actin is indicated below each immunoblot image. (**b**) The cells were treated with different doses of γ-radiation and the survival fraction was determined using clonogenic formation assay (*n* = 4). Cells were treated with γ-radiation of 10 Gy and dead cell populations were determined by flow cytometry after staining with Annexin V and PI (*n* = 3). (**c**) Representative image of PI-Annexin V double staining examined in lung cancer cells and (**d**) graph of dead cells is shown. Bars represent the means ± SEM. ** *p* < 0.01 and *** *p* < 0.001 by two-way analysis of variance.

**Figure 3 cancers-11-00386-f003:**
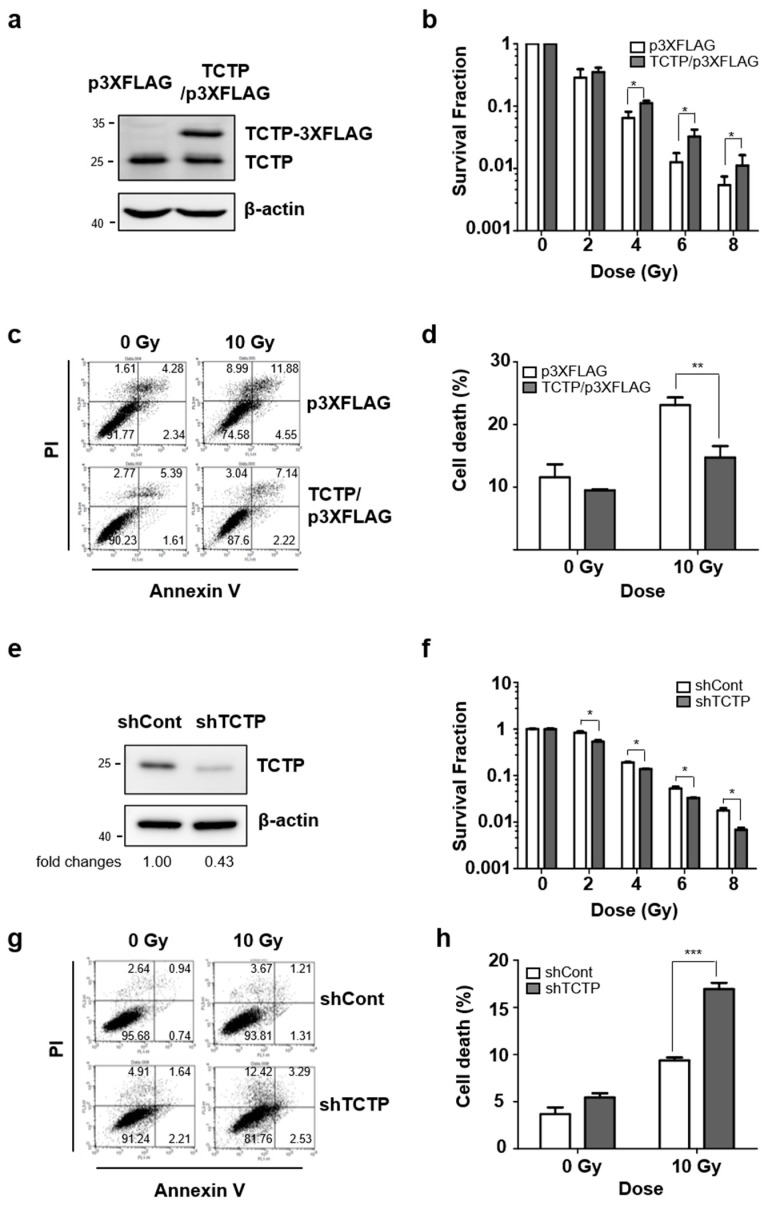
TCTP involves in radioresistance in cancer cells. (**a**) MCF7 cells were overexpressed with TCTP-3Xflag and overexpression of TCTP-3Xflag was confirmed using Western blot analysis. (**b**) TCTP-3Xflag-overexpressed MCF7 cells were treated with different doses of γ-radiation and the survival fraction was measured by clonogenic formation assay (*n* = 3). Cells were treated with γ-radiation of 10 Gy and dead cell populations were determined after 48 h. (**c**) The representative images of PI-Annexin V double staining examined in TCTP-overexpressed MCF7 cells and (**d**) the graph of apoptotic cells is shown (*n* = 3). (**e**) A549 cells were transfected with TCTP shRNA (shTCTP) and control shRNA (shCont) and TCTP down-regulation was confirmed using Western blot analysis. (**f**) TCTP-knockdown A549 cells were treated with different doses of γ-radiation and the survival fraction was measured using clonogenic formation assay (*n* = 6). Cells were treated with γ-radiation of 10 Gy and cell death was analyzed using flow cytometry (*n* = 3). (**g**) The representative image of PI-Annexin V double staining examined in TCTP-knockdown A549 cells and (**h**) the graph of the dead cell population is shown. The cropped blots are used in the figure, and full-length blots are presented in Appendix A. Bars represent the means ± SEM. * *p* < 0.05, ** *p* < 0.01, *** *p* < 0.001 by two-way analysis of variance.

**Figure 4 cancers-11-00386-f004:**
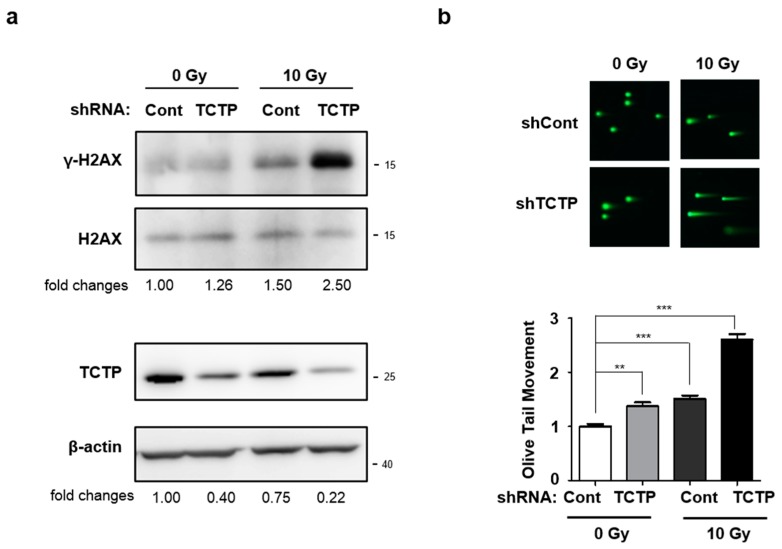
TCTP knockdown potentiates radiation-induced DNA damage in A549 cells. (**a**) After 48 h γ-radiation, γ-H2AX, H2AX, and TCTP expression level of A549 cells transfected with TCTP shRNA (shTCTP) and control shRNA (shCont) were measured using Western blot analysis (*n* = 4). The cropped blots are used in the figure and full-length blots are presented in Appendix A. The relative level of TCTP to β-actin and γ-H2AX to H2AX are indicated below each immunoblot image. (**b**) TCTP-depleted A549 cells were treated with 10 Gy of γ-radiation and comet assay was performed after 48 h. The olive tail moment was calculated using the Comet 5.5 software. The data represent the mean ± SEM. ** *p* < 0.01, *** *p* < 0.001 by one-way analysis of variance.

**Figure 5 cancers-11-00386-f005:**
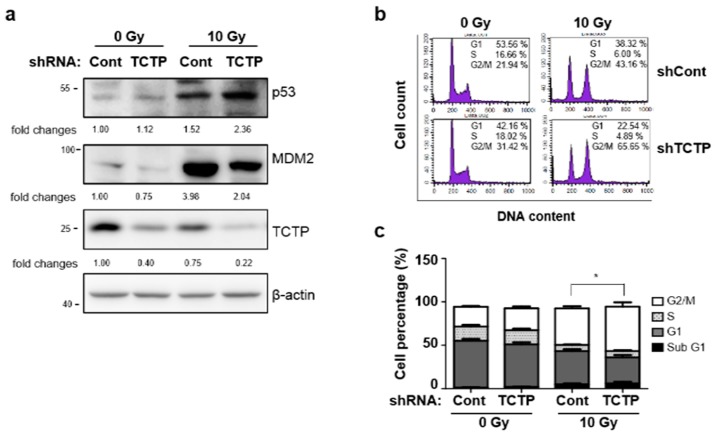
p53 is involved in TCTP-mediated radioresistance of A549 cells. (**a**) A549 cells were transfected with control shRNA (shCont) or TCTP shRNA (shTCTP) and treated with 10 Gy of γ-radiation. Protein lysates were prepared and analyzed by Western blotting 48 h after radiation treatment. The relative level of proteins in comparison to β-actin is indicated below each immunoblot. (*n* = 5) (**b**) The representative image of cell cycle analysis in shCont and shTCTP A549 cells is shown. Cells were treated with 10 Gy of γ-radiation and analyzed 24 h after radiation. (**c**) Graph of each cell population is shown. * *p* < 0.05 by student’s *t*-test in comparison of G2/M phase. (**d**) The representative image of PI-Annexin V double staining examined in p53 siRNA and TCTP shRNA-transfected A549 cells is shown. (**e**) A549 cells were immunoblotted using anti-p53 antibody to confirm p53 knockdown. (**f**) A549 cells were transfected with indicated siRNA and exposed to 10 Gy of γ-radiation. Cell death was measured using Annexin V/PI assay 48 h after γ-radiation treatment. The cropped blots are used in the figure and full-length blots are presented in Appendix A. Bars represent the mean ± SEM (*n* = 3). * *p* < 0.05 and *** *p* < 0.001 by two-way analysis of variance.

**Figure 6 cancers-11-00386-f006:**
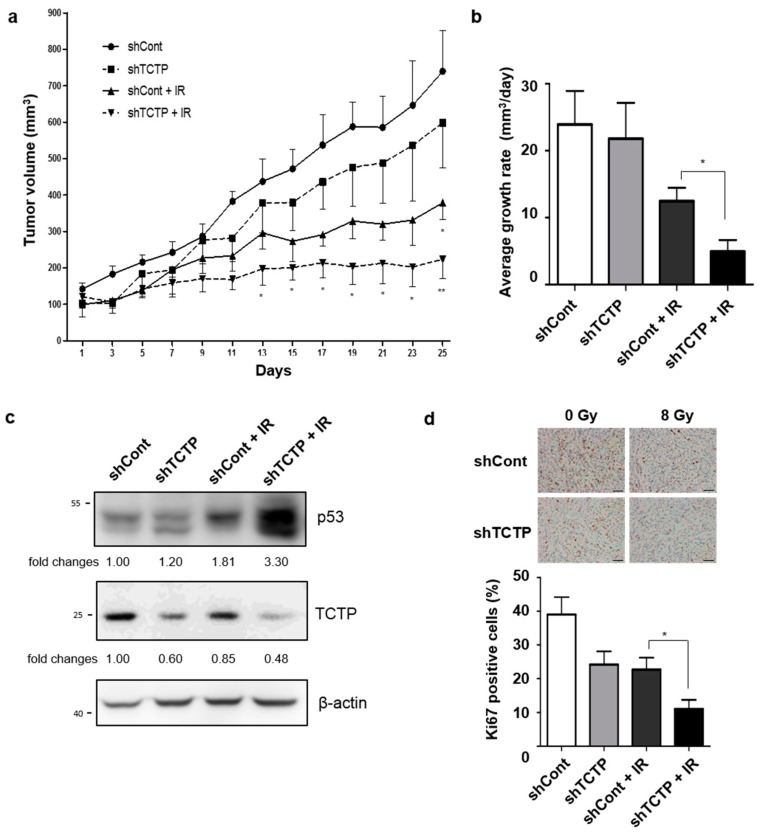
TCTP knockdown enhances radiosensitivity in tumor xenograft mouse model. A549 cells were stably infected with lentiviral control shRNA (shCont) and shRNA vector-targeting TCTP (shTCTP) and introduced to the right thigh of 6-week-old, female, Balb/c nude mice. When the tumors of stably transfected cells reached 100 mm^3^, they were treated with γ-radiation (8 Gy). (**a**) The growth of the tumors was assessed by measuring tumor volume with digital caliper every other day for 25 days. The data represent the mean ± SEM. * *p* < 0.05, ** *p* < 0.01 vs. shCont. (**b**) Average tumor growth rate (mm^3^/days) is shown. The data represent the mean ± SEM (*n* = 6–8/group). * *p* < 0.05 (**c**) The immunoblots of tumors from shCont, shTCTP, shCont+IR, and shTCTP+IR groups using anti-p53, anti-TCTP, and anti-β-actin antibodies are shown. The cropped blots are used in the figure and full-length blots are presented in Appendix A. The relative level of proteins in comparison to β-actin are indicated below each immunoblot (*n* = 6). (**d**) Immunohistochemistry of the tumors with anti-Ki67 antibody and the percentage of Ki67-positive cells per field are shown. The scale bar indicates 50 μm (magnification: 200×). The data represent the mean ± SEM. * *p* < 0.05.

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
