# Peer review of "Radiosensitivity of Cancer Cells Is Regulated by Translationally Controlled Tumor Protein"

_cancers, 2019, doi:10.3390/cancers11030386_

Round 1
Reviewer 1 Report
This is a well written and done study these are few suggestions authors can consider
Since there are many freely available databases with patient gene expression, could authors shown lung cancer patient correlation to TCTP. I think this will strengthen the observation of authors.
P53 increases pro-apoptotic proteins and p21, leading apoptosis and cell cycle arrest. Authors have shown apoptosis changes, since TCTP acts via p53, can authors investigate or comment on cell cycle arrest.
p53 is activated by by ATM/ATR kinases following DNA damage, authors said the data not shown, but that will be useful information to show how TCTP is regulated by ATM/ATR.
Author Response
Comments and Suggestions for Authors
This is a well written and done study these are few suggestions authors can consider

Reviewer 2 Report
In this manuscript, the authors demonstrate that the expression of Translationally controlled tumor protein (TCTP) determines the radiosensitivity of cancer cells in a p53-dependent manner. By modulating the expression of TCTP in a panel of radiation treated breast and lung cancer cell lines, the authors claim that TCTP protects cells from DNA-damage and promotes p53 degradation, thereby reducing cell death in response to radiotherapy. The manuscript is well-written and is acceptable for publication provided the following issues are appropriately addressed.
- Representative images of the clonogenic assays could be included in all the relevant figures.
- In Fig. 1a and 2a, p53 westerns post-radiation treatment could be performed to test if p53 status of these cell lines determines the RT response. In addition, the p53 status of all the cell lines should be discussed as to how it may relate to radioresistance by TCTP. E.g H1299 cells are p53 null while A549 and H460 are p53 wild-type and yet the authors observed a differential response to radiation in terms of cell death and clonogenic survival.
- In Fig. 2c., why do the FACS plots for H1299 and H460 cell lines do not match with the quantitation shown in 2d? It appears that the percent apoptotic population in these cells in lower than in A549 cells. Apoptotic population is only the cells that fall in the lower right quadrant. The upper quadrants have necrotic (UR) and dead cells (UL) and should not be pooled with the apoptotic population.
- Typically, at 10 Gy dose, the apoptotic response is p53 independent. Did the authors try a lower dose such as 5 Gy in experiments shown in Fig 5 to more rigorously test the p53 dependence of TCTP mediated radioresistance?
Author Response
Point 1: Representative images of the clonogenic assays could be included in all the relevant figures.
Response: As suggested by the reviewer, we included all the currently available representative images of the clonogenic assays in Figure 1b and Figure S2.
Point 2: In Fig. 1a and 2a, p53 westerns post-radiation treatment could be performed to test if p53 status of these cell lines determines the RT response. In addition, the p53 status of all the cell lines should be discussed as to how it may relate to radioresistance by TCTP. E.g H1299 cells are p53 null while A549 and H460 are p53 wild-type and yet the authors observed a differential response to radiation in terms of cell death and clonogenic survival.
Response: Thank you for your valuable comment. As the reviewer has mentioned, the p53 status of cancer cell lines used in this research are different. A549, H460 and MCF7 cells have p53 wild-type; whereas, H1299 cell is p53 null cancer cell line, and T47D and MDA-MB-231 cells are p53 mutant cancer cell lines. In our research we have shown that TCTP knocked down A549 cells showed increase in p53 level and p53 involvement in TCTP-mediated radioresistance. We have not performed experiments in p53 involvement in TCTP-mediated radioresistance in other cell lines, but we can find some clues to the mechanism from previous researches.
TCTP is one of genes that are directly repressed by wild type p53. Also TCTP is known to promote degradation of wild type p53 by associating with MDM2 thereby suppressing MDM2 auto-ubiquitination (Amson et al. Nat Med 2011). Induction of wild type p53 in H1299 cells, HCT116 cells, and thymus and spleen of Trp53+/+ mice all show decreased intracellular TCTP level, whereas modulation of TCTP level in HCT116 and H1299 cells induced reciprocal regulation of wild type p53 level. In this context, it is important to note that p53 mutation is more common in TCTP-rich breast cancer, suggesting that TCTP repression function of wild type p53 is lost in mutant or null p53 (Amson et al. Nat Med 2011). The increased TCTP level is known to cause malignancy in cancer cells (Nagano-Ito et al. Biochem Res Int 2012), and TCTP itself is known to give protection against DNA damage (Zhang et al. PNAS 2012 and Li et al. Oncogene, 2017). Thus it can be suggested that p53 mutation can induce more TCTP expression, increasing radioresistance in cancer cells.
Point 3: In Fig. 2c., why do the FACS plots for H1299 and H460 cell lines do not match with the quantitation shown in 2d? It appears that the percent apoptotic population in these cells in lower than in A549 cells. Apoptotic population is only the cells that fall in the lower right quadrant. The upper quadrants have necrotic (UR) and dead cells (UL) and should not be pooled with the apoptotic population.
Response: We appreciate reviewer's detailed evaluation of our manuscript. All of our quantification of AnnexinV/PI analysis results is based on dead cell population (AnnexinV+PI-, AnnexinV+PI+, and AnnexinV-PI+), and thus we addressed γ-radiation induced cell death instead of using apoptosis. We found that we described the results as apoptosis in legends of figure 1 to 3 and corrected them to dead cell population.
Point 4: Typically, at 10 Gy dose, the apoptotic response is p53 independent. Did the authors try a lower dose such as 5 Gy in experiments shown in Fig 5 to more rigorously test the p53 dependence of TCTP mediated radioresistance?
Response: We appreciate the reviewer's comment. As the reviewer has suggested, it may have enriched our results if we have also performed experiments with lower dose such as 5 Gy to determined p53 involvement in TCTP-mediated radioresistance, but we believe that using 10 Gy of radiation is sufficient to suggest p53 involvement in TCTP-induced radioresistance. As shown in Fig. 5a, p53 is increased after the 10 Gy radiation in control shRNA transfected A549 cells, indicating the involvement of p53 in them cell death response. There are many research suggesting p53 involvement in gamma radiation at 10 Gy dose or higher dose (Hermeking et al., Mol Cell 1997; Jeremy E et al., Science 2012; Roberta et al., Nat Cell Biol 2000; Katsutoshi et al., Cell 2000). Also a research suggest dual effect of p53 in radiation treatment. The p53 null mice showed increase in cell survival compared to control at radiation dose below 12.5 Gy whereas, radiation above 12.5 Gy caused increase in sensitivity of p53 null mice suggesting p53-dependent apoptotic response in gamma radiation below 12.5 Gy and p53-independent apoptotic response in gamma radiation above 12.5 Gy (Komarova et al., Oncogene 2004). Thus we believe that using 10 Gy was adequate in this research.
Reviewer 3 Report
In this study, the authors investigated the involvement of translationally controlled tumor protein (TCTP) in radiosensitivity of breast and lung cancer cells. The present study showed that overexpression of TCTP resulted in the resistance of cancer cells to ionizing radiation, while inhibition of TCTP expression increased the radiosenstivity of cancer cell. In addition, the present study showed that p53 mediates the regulation of radiosensitivity by TCTP. This study is well-performed and well-written. Furthermore, the topic of this study is important and the obtained results are very interesting. On the whole, in my opinion, this manuscript is suitable for Cancers.
Minor comments: The image quality of each Figure should be improved.
Author Response
Minor comments: The image quality of each Figure should be improved.
Response: As reviewer suggested, we exchanged each figure with improved resolution.
Round 2
Reviewer 2 Report
The authors have addressed the comments appropriately and the manuscript is now acceptable for publication.